# Public Libraries and Walkable Neighborhoods

**DOI:** 10.3390/ijerph16101780

**Published:** 2019-05-20

**Authors:** Noah Lenstra, Jenny Carlos

**Affiliations:** Library and Information Science Department, University of North Carolina at Greensboro, Greensboro, NC 27402, USA; jjcarlos@uncg.edu

**Keywords:** public libraries, walking programs, public health partnerships, public programming, librarianship, library and information science

## Abstract

Public libraries constitute a ubiquitous social infrastructure found in nearly every community in the United States and Canada. The hypothesis of this study is that public libraries can be understood as important supports of walking in neighborhoods, not only as walkable destinations, but also as providers of programs that increase walking in communities. Recent work by public health scholars has analyzed how libraries contribute to community health. This particular topic has not previously been researched. As such, a qualitative, exploratory approach guides this study. Grounded theory techniques are used in a content analysis of a corpus of 94 online articles documenting this phenomenon. Results show that across North America public librarians endeavor to support walking through programs oriented around stories, books, and local history, as well as through walking groups and community partnerships. While this exploratory study has many limitations, it does set the stage for future, more rigorous research on the contributions public libraries and public librarians make to walking in neighborhoods. The principal conclusion of this study is that additional research is needed to comprehensively understand the intersection between public librarianship and public health.

## 1. Introduction

Since 2014, Misty Von Behren, Deputy Director of the Perry Public Library in the US state of Iowa, has led a walking school bus program in her small town of 7702. On Fridays throughout the year, 50 to 70 elementary school aged children walk from the town’s high school to the elementary school, led by the public librarian and other volunteers that she coordinates. The program has been so successful that in 2017 Von Behren was asked to speak about it at a state conference for librarians, to encourage others to lead similar initiatives [1,2,3].

The purpose of this article is to explore how public libraries, and public librarians like Von Behren, contribute to walking in neighborhoods. Since this topic has received very little research, the methods used are primarily qualitative. The hypothesis of this study is that public libraries can be understood as important supports of walking in neighborhoods, not only as walkable destinations, but also as providers of programs that increase walking in communities.

As more and more information becomes available online and accessed through digital devices, public libraries have in some cases dramatically re-thought their roles in communities [4]. Librarians have shifted attention away from collections and reference, and towards public programs and events, offered both in the library building and off-site. The American Library Association states that “as US libraries transform… public programming is rising to the forefront of daily operations” [5]. A 2018 systematic study of voter perceptions of public libraries finds “significantly more voters today (43%) describe the library as a place that ‘offers activities and entertainment you can’t find anywhere else in the community,’ than did in 2008 (34%), and more believe this is an important role for a library (48% in 2018 vs. 38% in 2008)” [6]. More and more public libraries offer engaging programs and events, only some of which have to do with literature, literacy or reading. These new roles are increasingly expected of the taxpayers who fund libraries.

In the public health literature on walkable communities, public libraries are framed as destinations that should be accessible by walking. In spring 2018 the United States (US) Centers for Disease Control and Prevention (CDC) unveiled the “Connecting Routes + Destinations” campaign to increase physical activity. The campaign focuses on making it easier for people to walk, bike, and use a wheelchair throughout their communities [7]. Public libraries—alongside schools, worksites, homes, grocery stores, and parks—are framed as key destinations to which walkable routes should be made (Figure 1). The context for this decision was that in the United States public libraries are a vital social infrastructure present in almost every community in the country. As of spring 2017 there are more public libraries in the US than either McDonald’s or Starbucks [8].

Given this fact, over the past five years public health researchers and policy makers have attended to the public health roles of public libraries. The American Public Health Association states that the fourth most read public health news story in their newsletter during 2018 was the article “Libraries, public health work together on community health” [9]. Public health scholars also frame public libraries as vital contributors to health and wellbeing. Program officers from the Robert Wood Johnson Foundation and RAND Health see the presence of a public library as a measure of a community’s health [10]. Public health scholars from New York’s Columbia University frame public libraries as “a community-level resource to advance population health” [11]. Faculty from the University of Pennsylvania’s Family Medicine and Community Health program have since 2015 worked on developing what they call the “Healthy Library Initiative,” an evidence-based campaign to “to establish the feasibility of partnering with public libraries to improve population health” [12].

This recent interest in public libraries in the public health sector builds upon decades of research by Library and Information Science (LIS) researchers on how public libraries contribute to health [13]. Collaborations between public libraries and health professionals abound in this literature: Rubenstein found in her archival research that in the 1940s a bookmobile that travelled throughout the rural, Appalachian region of Georgia carried a nurse who provided communities with vital healthcare services to which they would not otherwise have access [13]. In the 1970s, the sub-field of consumer health information services developed in public libraries throughout parts of the nation [14]. More recently, the range of health programs offered in public libraries has dramatically increased, and now includes everything from healthy living support groups to chair-based exercise classes for older adults to healthy cooking and home repair classes [15,16]. Recognizing the rapid growth of this area, the US Public Library Association released in May 2019 a new suite of tools designed to help public librarians assess the impacts of their health programs and classes [17].

Past work on how public libraries contribute to healthy communities has focused on what takes place within the building of the public library. However, a significant amount of public library programs take place outside the library building, through outreach and other services. The classic image of the bookmobile encapsulates the ideal of library services being broadly available throughout a community, and not just in one location [18].

This article adds to this literature by exploring how public librarians, through programs they develop that take place outside the library building, support walking in neighborhoods. The focus is primarily on North America and the United Kingdom, and the results are tentative and exploratory. Nonetheless, there is evidence that librarians indeed support increased walking in neighborhoods. The ultimate goal of this article is to foster conversations and dialogue in the public health sector about how public libraries can contribute to campaigns to foster and support walkable neighborhoods in innovative ways not previously discussed or analyzed in the research literature.

## 2. Materials and Methods

To analyze this topic, two previous studies are drawn upon. In spring 2017, a self-selecting sample of 1157 public libraries in the US and Canada (or 5.9% of the 19,564 public library locations spread across these two nations [19,20]) filled out an online survey about how their libraries support active living through public programs [21]. Public libraries throughout North America were invited to self-select for participation in the survey. Data collection was carried out via an online questionnaire using Qualtrics. The URL to the questionnaire was sent to public librarians in the US and Canada through state and provincial library electronic mailing lists, as well as through announcements from state and provincial libraries to public libraries in their regions. In addition, the survey was disseminated through national electronic mailing lists used by public librarians and on the project’s website. To ensure that a given library only filled out the survey once, respondents were asked to include the postal code of their library. This data enabled removing any redundant responses. The postal codes also enabled sorting the respondents into the four-part community classification system used by the federal government in the United States [22] (Table 1).

The use of self-selecting web surveys as a research method to better understand poorly researched topics is well documented. An example of this is the Great British Class Survey: “a web survey that ran on the BBC website from January 2011 to July 2013 and elicited 325,000 responses” [23]. While by no means providing a representative sample of the population surveyed, this means of data collection offers “an unrivalled opportunity to explore” [23] poorly understood phenomenon. Since the overall orientation of this study is exploratory and qualitative, this method is a good way to begin to develop our understanding of connections between walkable neighborhoods and public libraries.

The survey included a series of questions on walking programs, and those data are analyzed here. In particular, the survey asked “Has your library ever offered any programs or services that include Walking, hiking, bicycling, or running” and “Has your library ever offered any programs or services that include StoryWalks?” The full survey is included in Appendix A. See below for more information on what a StoryWalk entails. To focus only on outdoor programs, analysis was restricted to those libraries that indicated their walking programs took place “outside the library” (Question #12 on the survey). This restriction is necessary to differentiate between programs that feature a speaker talking about walking from programs that focus on community members actually walking outside in neighborhoods. The only demographic data the survey collected was postal code, which enabled the deletion of redundant responses and the sorting of respondents along a rural-urban continuum using data from the US Institute of Museum & Library Services [22], Table 1.

Next, to better understand what these walking programs entail, this article draws upon a corpus of online publications collected on active living programs in public libraries. Since spring 2017, the lead author has had daily Google News Alerts set up to collect anything published on the internet that uses the terms “walk” + “public library”; “walking” + “public library”; “story walk” + “public library”; or “storywalk” + “public library.” Articles that feature substantive discussion of walking programs, as opposed to the mere mention of a program’s existence, were added to the online citation management system Zotero, which also collects full-text copies of online publications. The articles collected through these methods were supplemented by additional articles identified through more traditional scholarly search techniques, in particular searching the databases Web of Science, Scopus, LISA, LISTA, DOAJ, Google Scholar, and WorldCat. This searching revealed only three research articles on this phenomenon, all three of which consist of action research case studies that where limited to a particular community (Table 2). The most common sources were the media, trade publications written by librarians for other librarians, and online blogs. In spring 2019, the authors analyzed the 94 articles in this corpus (see Appendix B).

Standard techniques from content analysis [24] were used to analyze the data. In particular, the method of “problem-driven analysis” was utilized. This sort of analysis “derives from epistemic questions, from a desire to know something currently inaccessible and the belief that a systematic reading of potentially available texts and other data could provide answers” [25]. The particular methods used derive from grounded theory [26]. These methods enable the construction of a research-based portrayal of how public libraries support walking in neighborhoods. In addition to using the coding and tagging features within Zotero, the researchers used spreadsheet software to closely analyze and qualitatively code the text of the articles in this corpus. Through the analysis of these two datasets, an empirically-derived portrait emerges of how public libraries support walking in neighborhoods. This portrait is incomplete, and has many limitations, discussed below. Nonetheless, being the first research on this topic, this article sets the stage for more future, more rigorous investigations.

## 3. Results

Figure 2 displays the locations of the 482 public libraries across North America (41.7% of all respondents) that indicated in the survey they had offered walking, hiking, bicycling, or running programs that took place outside of the library building. More specifically, 281 (24.3%) indicated they had offered general walking, hiking, bicycling, or running programs and 303 (26.2%) indicated that they had offered StoryWalk programs (respondents were asked to indicate all that applied). As a self-selecting web survey, this survey by no means provides a representative sample of public libraries. Nevertheless, Figure 2 and Table 1 do suggest that these phenomena are geographically widespread throughout North America.

The one dot in Figure 2 in far northern Alaska is not an error. That dot represents the Tuzzy Consortium Library, which serves the mostly indigenous community of Utqiagvik. Every Summer the library offers a series of “Lagoon Walks” programs. These programs emerged from a partnership among “the regional diabetes program, the library and the local science community [and they include] walking, science lectures, literature and a weekly two-mile walk aimed at increasing activity in the community” [27]. All walks begin and end at the library, and are hosted and sponsored by librarians. This example illustrates the broad and deep reach that public libraries have in diverse communities across North America.

This survey data illustrates that at least some public libraries throughout North America offer outdoor walking programs. To better understand these programs, this article now turns to the second study, which consisted of a qualitative analysis of online publications related to this trend. This analysis revealed that public libraries support walking in neighborhoods in four different ways:The library as a resource for stories: Connecting stories and walkingThe library as a community center: Walking programs without storiesThe library as a community partner: Walking through partnershipsThe library as a walkable destination: Improving walking routes to libraries

### 3.1. The Library as a Resource for Stories: Connecting Stories and Walking

There are three ways libraries connect stories and walking in outdoor programs:StoryWalksHeritage walksWalking book clubs

#### 3.1.1. StoryWalks

Among the 94 articles analyzed, 45 (48%) focused on StoryWalk programs, suggesting this type of program is the most (or among the most) common type of walking program that public libraries develop. The StoryWalk concept was developed by Anne Ferguson and the public library in Montpelier, Vermont in the late 2000s. A StoryWalk consists of a deconstructed children’s book posted along a walking trail, typically in a park, but also sometimes in downtown business corridors or along greenways. Families then walk and read at the same time [28]. In Lewes, Delaware, and Boone, North Carolina, downtown businesses agreed to post different pages of a StoryWalk in their store windows [29,30]. In Boston, a branch of the public library system created a StoryWalk spread throughout the Latin Quarter of the city to celebrate Latinx heritage month [31]. The concept has spread throughout the United States and beyond, with the Vermont public library where it originated reporting that StoryWalks have been installed in 50 states and 12 countries [28].

StoryWalks can be either temporary or permanent. Initially conceived of as temporary installations that would be mounted the same way that temporary political advertisements are mounted, over time a growing number of libraries have created permanent, weather-proof StoryWalk installations. These permanent installations can be opened from behind to change the stories on a periodic basis. In Burke County, North Carolina, the library set up a temporary StoryWalk along a greenway, which was so successful that in 2018 the library installed a permanent StoryWalk installation at the same location. Library director Jim Wilson said the StoryWalk represents “a unique and wonderful opportunity for local families to not only read and discover the wonderful stories found in books, but also walk a quarter mile on our beautiful greenway” [32]. Evidence of how embedded the StoryWalk concept has become in the public library profession is the fact that at least two vendors now specialize in selling these permanent StoryWalk installations to librarians [33,34]. At least one library has hired a StoryWalk project manager to organize these programs on an ongoing basis [35].

These programs have two purposes: To get families walking together and to get families reading together. Describing a StoryWalk organized by the Rochester Public Library in Minnesota, a program manager stated “It’s a good way to get families out and to see another part of downtown” [36]. According to the US Association for Library Service to Children, StoryWalks are ideal outreach activities since they increase visibility of the library [37]. Tracy Horvath, library director of the Mt. Juliet Library in Tennessee, stated “I’m for any way to open the world of books to children, and if reading is combined with a healthy stroll, what could be better?” [38]. Some librarians also frame StoryWalks as efforts that explicitly contribute to building more walkable neighborhoods. For instance, librarians from St. Mary’s County Library in rural southern Maryland state that through their StoryWalks “we provide access to built environments that encourage more physical activity” [39].

#### 3.1.2. Heritage Walks

A second way libraries connect stories and walking is heritage walk programs. These programs involve librarians weaving together local history with walking. In 2017, the American Library Association highlighted the spread of these programs throughout the country [40]. A branch of the Baltimore, Maryland library hosts regular historic walking tours, and the library in La Crosse, Wisconsin, does ghost walking tours every fall [41,42]. In New Jersey, the Red Bank Public Library celebrated its 80th anniversary by conducting a walking tour of the town, and in Effingham, Illinois, the library helps organizes an annual sculpture walk [43,44].

These programs also take place outside the US The public library in Innerpeffray, Scotland, launched a heritage walk to showcase its local history [45]. Walking history programs have been a major initiative at the library in Winnipeg, Canada, where the library regularly offers “historical walks, literary walks, gardening walks, and walking book clubs” [46] to expose people to the history and resources of the city that can be accessed through walking.

#### 3.1.3. Walking Book Clubs

A third way libraries connect stories and walking is through walking book clubs. It is unknown when this concept first emerged, but it has been offered for at least a decade. In the late 2000s the public library in Elgin, Illinois launched its walking book club. According to the librarian that organizes the program “we read and exercise, build community and friendships, and hold lively discussions along the way” [47]. One participant said that “the friends I have made will last me longer than the books I have read” [48].

In Kokomo, Indiana, participants walk around downtown, discussing what they are reading [49]. In Middletown, Connecticut, Book Talk with a Walk programs are held Mondays from noon to 1:30 pm Participants walk from the library to the river and back, two miles round-trip. Librarian Ann Smith said “Even if you haven’t read the book, everybody has something to contribute because you have an opinion about… whatever the book’s theme was: everybody’s got something to offer, whether it’s related to the book or not” [50]. This quote illustrates how these book clubs tend not to be overly focused on the books themselves, and indeed in some places the literary dimensions of these clubs are entirely absent, as will be discussed below.

### 3.2. The Library as a Community Center: Walking Programs without Stories

Libraries also promote walking in neighborhoods by developing walking programs without literary dimensions. These programs build upon the concept of the library as a community center [4]. Public libraries in Fife, Scotland, developed the Library Walk ON program in the mid-2010s. From 2017 to 2018, 113 walks with 1187 participants took place across the 34 public libraries in this library system located north of Edinburgh, and 25 library staff have been trained as health walk leaders, with some also receiving additional training as Dementia Friendly Walk leaders. The library also offers Buggy Walks for individuals with strollers. According to organizers, “walks progress at the pace of the least able walker and since some group members use mobility aids, it’s important each walking route is planned to make sure they are accessible” [51]. The Walk ON program explicitly focuses on increasing walkable neighborhoods. A librarian stated that “we’ve aimed to change attitudes towards walking by showing the variety of easily accessible walk routes in our communities,” and the program “encourage[s] regular walking and reduced car use” [51].

A similar library walking initiative emerged in the Pittsburgh, Pennsylvania, area in 2008. There, the Allegheny County Library Association helps their 46 member libraries offer Wise Walk programs [52]. According to the Carnegie Library of Pittsburgh, a Wise Walk is a librarian-led 1–2 mile walk around the neighborhood geared toward those aged 50 and older [53]. While comprehensive statistics are not available, one participating library reports that over the years, registration varied from 10 to 30 in this ten-week program offered twice a year in the spring and fall [54].

Other librarians develop walking programs by themselves, without a larger structure like the Walk ON or Wise Walk program model to follow. In Arkansas, staff from the Dee Brown Public Library decided to lead walking programs in an adjacent park. They found that prior to these programs “walkers often tell me [the librarian] they didn’t know it [the park] was here until they joined our walking group” [55]. In Maitland, Florida, librarians developed several measured trails that originate from the library for its weekly Maitland Walks club [56].

Sometimes these programs have ancillary purposes beyond walking. In Manitowoc County, Wisconsin, the library organizes an annual Library to Library Fun Walk, in which participants walk from one county library to another county library [57]. Here one of the goals is to increase awareness of the different libraries in the county. Other libraries organize walking programs with environmental components. In Massachusetts, the Oak Bluffs Public Library organized a Community Walk and Clean-Up day in which members of the community picked up trash as they walked around the neighborhood around the library [58]. Librarian Carolina Cooney said she intends to make this a monthly program. Still others combine bookmobile services with walking. In Lethbridge, Alberta, the bookmobile is the start point for a walking club where patrons can borrow walking poles to use from the library if they need them [59]. The library in Helena, Montana, hosts a monthly walking group when the bookmobile visits a rural park [60].

Why do librarians develop these programs? One answer to this question comes from Eureka, Illinois, where library director Ann Reeves said that she developed the Roaming Readers Walking Club, which meets weekly at the library for a 30-min walk, because “in a larger community, there might be better options (for a public fitness program) … but we don’t have a community center. We are the community center. If we can provide these services, it helps us and it helps the town” [61]. This quote illustrates the idea discussed in the introduction, namely that Americans increasingly expect public libraries to fill in gaps in social services, providing things that otherwise would be inaccessible in local communities [6].

### 3.3. Participate in Community Partnerships: The Library as a Community Partner

A third way that libraries support walking in neighborhoods is by participating in community partnerships. These partnerships can take many different forms, depending on the constellation of partners involved in them. These partnerships sometimes involve collaborations with researchers. In Kingston, Ontario, public libraries collaborated with Kinesiology researchers from a local university for a pedometer lending project whose purpose was to examine the feasibility of lending pedometers to library patrons to increase walking [62]. The researchers found that 92.1% of participants agreed that the pedometer “acted as a motivational tool to inspire an increase in walking” and 78.9% agreed that “the pedometer succeeded in motivating them to set a walking goal” [62]. A similar researcher–librarian collaboration emerged in Alberta. There, researchers developed a ‘Pedometer Library Loan Program’ (PLLP) with local libraries [63].

In North Carolina, the Farmville Public Library partnered with a researcher from the University of North Carolina at Chapel Hill for a Get Walking at Your Library program. This program involved circulating pedometers from the library and having participants fill out a health assessment tool when they returned the devices [64]. This collaboration inspired the librarian to take a more active role in promoting walking. After the collaboration ended, the librarian partnered with the local Parks & Recreation Department to develop a Farmville Moves program in which participants train together to complete their first 5 kilometer race together [65]. The library provides health and exercise books, as well as health and wellness professionals who give short talks about staying healthy and exercising safely.

Other partnerships in other places involve: walking and running groups, health professionals, and urban planners. In Ohio, the Worch Memorial Library partnered with a local Volkssport Association chapter for a walking event [66]. In Sacramento, California, librarians partnered with Downtown Grid Sacramento: The library distributed walking maps of downtown and the library’s book club created a guided tour of downtown to visit those places [67]. Partnerships can also include national organizations. Librarians in Colorado, Texas, and Florida have formed partnerships with chapters of national non-profit Walk With a Doc to offer programs in which patrons walk with doctors and other medical professionals [68,69,70].

These partnerships are not restricted to urban areas. As discussed above, in the rural, primarily indigenous community of Utqiagvik in far northern Alaska, public librarians partner with a variety of entities to offer “summer lagoon walks” every year. Youth participants go walking with local experts, who educate them about the ecology, culture, and environment of the community as they walk together [71]. In the small town of Woodstock, New Brunswick, librarians partner with a local running club so that the club hosts their monthly meetings at the library. The library provides the running club with space and access to running books and periodicals, and the running group encourages patrons of the library to join them on their fun runs, which start and end at the library. According to the librarian, “no one gets left behind, and all ages and genders turn out, from parents with their kiddos in strollers to some slow-moving elderly walkers that come 15 min early to get a head-start” [72].

Finally, the library in the small town of Irvine, Kentucky partners with its local hospital and health department for a variety of walking programs. One of these is a walking club that meets Tuesday evening for a 30 min walk of downtown Irvine. Devices are available to record resting and active heart rates before and after the walk. Given that the library serves a rural area, the library also encourages bookmobile patrons who cannot make it to the library to use a pedometer or step counting app to track steps [73]. The library also works with partners to turn Irvine into a Kentucky Trail Town by working to increase trails and greenways throughout the county.

### 3.4. Improve Walking Routes to Libraries: The Library as a Walkable Destination

Some librarians also work with partners to increase safe walking routes to their libraries. At least two libraries (Galion, Ohio and Seymour, Indiana) participate in the national Walk [Your City] campaign [74,75]. In these communities, librarians and others installed signs throughout town indicating how long it takes to walk to the library from different points. In Arkansas, the Springdale library participated in the town’s Safe Streets planning commission, with the commission’s pilot project focused on linking “Springdale’s revitalizing downtown area with the hospital and public library,” such that one could walk or bike easily among these destinations [76]. In San Antonio, Texas, the public library developed two walking paths on its land. Librarians stated that “library patrons visiting these libraries often express their appreciation of offering healthy exercise on the library grounds [77]. A recent renovation of the public library in Wilmette, Illinois, included the installation of ADA accessible walking pathways around the library [78].

Elsewhere, librarians create guides of walkable areas near the library. At the Maitland Library in Florida librarians organized walking trails throughout the community, which originate at the library [56]. In High Point, North Carolina, the library, in partnership with the Parks and Recreation department, received a Community Change grant from the national organization America Walks to create maps of walkable and safe routes throughout the city, which will be distributed at library events and utilized for library programs [79].

Finally, library programs focus on raising awareness of the importance of walkable neighborhoods. In April 2018, the St. Louis County Library in Missouri offered Walkability of your Neighborhood programs. These programs were offered in partnership with two non-profit organizations, St. Louis County Older Resident Programs (CORP) and OASIS. The program was offered at three different branches, and consisted of a one mile guided walk with a representative from CORP. During the walk, the sidewalks and surrounding areas were assessed for pedestrian safety. Patrons filled out their walkability assessment forms, which were provided by CORP. At the end of the walk, everyone met back in the branch to discuss findings and how to improve pedestrian safety around the library [80].

## 4. Discussion

The hypothesis of this study was that public libraries can be understood as important supports of walking in neighborhoods, not only as walkable destinations, but also as providers of programs that increase walking in communities. The findings presented illustrate that this work is indeed happening, in various ways, in communities throughout the US, Canada, and Scotland. As an exploratory study of an emerging topic, this article seeks to foster additional research rather than present definitive findings. We need a better understanding of the spread and impact of this trend.

The major limitation of this study derives from its exploratory nature. At the beginning of this study, virtually nothing was known about the connection between walkable neighborhoods and public libraries, beyond the fact that public libraries should be considered a walkable destination [7]. This gap was first addressed by inviting a self-selecting sample of North American public librarians to share whether or not they offer programs that include walking, and, second, by constructing and analyzing a corpus of texts that document the spread and contours of this trend. While by no means providing a representative sample of either the North American libraries surveyed nor the English-language online literature collected, this research has opened up new avenues for inquiry.

In addition to the limitations of qualitative research, this study is also limited in the fact that it focuses on English-speaking countries, and primarily on the United States of America. Additional research on other parts of the world, which have very different traditions of public librarianship [4], is needed to better understand this topic at a global level. This study, despite its limitations, has opened up promising new areas of inquiry that should be further investigated by public health scholars interested in new opportunities to assess and support walkable neighborhoods.

This discussion now turns to additional research needed to better understand and support this emerging role of public libraries. First, as public health professionals and researchers increasingly look to public libraries as walkable destinations [7] and as supports of community health [9,10,11,12], it is important to understand what public librarians actually do. This article has shown that public librarians develop and offer programs that seek to engage communities at multiple levels, serving both the mind and the body in programs that weave together stories and walking. In addition, some libraries offer programs focused entirely on walking, without any literary dimensions. These programs are often supported by partnerships with local, regional, and national entities. At a practical level, much more work can be done within the public health sector to develop collaborations with public librarians [9]. There is evidence that these partnerships are already underway in parts of North America, but much more could be done.

Second, at the research level, much more work on the impacts of these programs is needed. This study revealed that some researchers are already working with public librarians to develop walking initiatives [62,63,64]. These action research studies do include assessment within them. However, we need much more research on the general impacts that result from public librarians developing walking programs. Stated as a research question: What do public librarians and their programs add to walkable neighborhoods? This study showed they are adding something. Now we need to understand in detail what that something is.

Third, as this role develops, public health researchers and practitioners could help public librarians understand accessibility issues associated with these programs. Very few articles mentioned efforts to attend to the needs of differently abled populations. As an emerging trend led by individuals who do not typically have any training in the accessibility of the outdoor built environment, these programs could greatly benefit from the insights of public health professionals with more expertise on the universal design of the built environment. At a practical level, things as simple as choosing walking routes that are accessible, providing aids like walking poles, and inviting feedback from the community as to how these programs could be made more inclusive, could all be done. Public health practitioners could help librarians achieve this goal of accessible neighborhoods for all.

## 5. Conclusions

This article has shown that throughout North America, and beyond, public librarians are by themselves and with partners developing new and innovative ways to support walking in neighborhoods. This study has shown that at least 482 public libraries in the US and Canada have offered programs that include outdoor walking experiences in the neighborhoods that surround libraries. Based on a qualitative review of online publications on this trend, these programs can be productively understood as connecting to four roles of the library: (1) The library as a resource for stories, (2) the library as a community center, (3) the library as a community partner, and (4) the library as a walkable destination. While much more work is needed to develop and understand this trend, this article has opened up new avenues for inquiry and for partnerships between those in the public health fields and those in the library sector.

## Figures and Tables

**Figure 1 ijerph-16-01780-f001:**
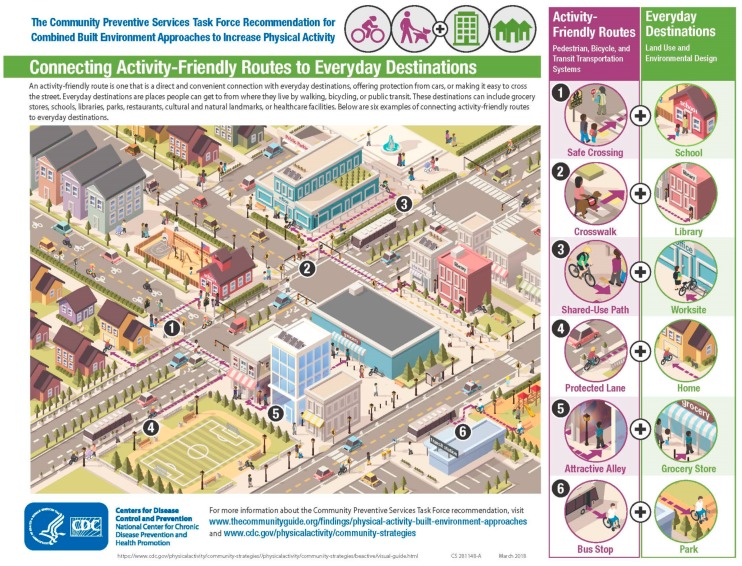
Connecting activity-friendly routes to everyday destinations. Image courtesy of Centers for Disease Control and Prevention (CDC) [7].

**Figure 2 ijerph-16-01780-f002:**
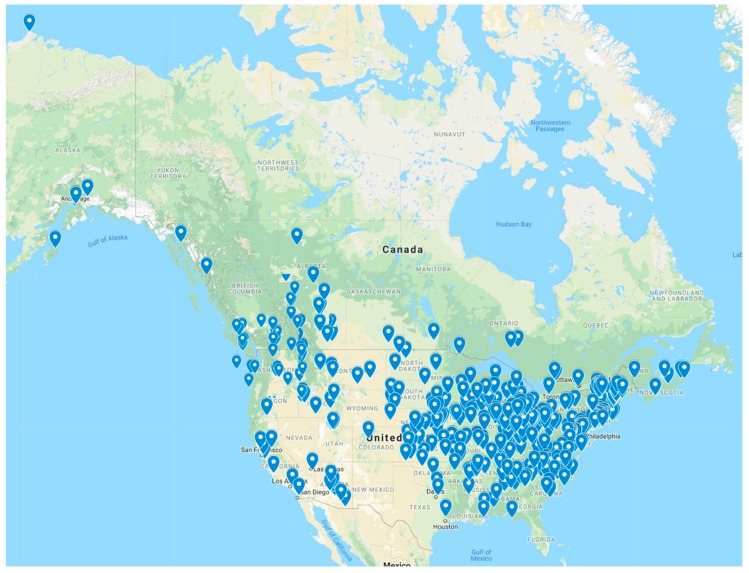
Map of public libraries in North America that indicated in an online survey that they have offered outdoor walking, hiking, bicycling, or running programs, *n* = 482.

**Table 1 ijerph-16-01780-t001:** Urban-rural classification of respondents, compared to break-down of public libraries in the United States. Source: [20]. Note: Comparable data from Canada is unavailable, see [19].

	Survey Respondents (*n* = 1157)	US Libraries (*n* = 16,568)	Offer Outdoor Walking Programs (*n* = 482)
**% Urban**	17.6%	17.4%	15.4%
**% Suburban**	36.2%	26.1%	34.9%
**% Town**	28.3%	20.2%	30.7%
**% Rural**	17.9%	36.4%	19.0%

**Table 2 ijerph-16-01780-t002:** Articles in online corpus, by publication type. See below for a description of items in table.

	Newspaper/Media Source	Trade Publication	Online Blog	Research-Based Source	Total
**StoryWalk**	31	7	7		45
**Heritage Walk**	7	2			9
**Walking Book Group**	3	4	2		9
**Walking Clubs**	5	5	3		13
**Walking Partnerships**	3	4	1	3	11
**Walkable Destination**	4	2	1		7
**Total**	53	24	14	3	94

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
