# Peer review of "Public Libraries and Walkable Neighborhoods"

_ijerph, 2019, doi:10.3390/ijerph16101780_

Round 1

Reviewer 1 Report

This paper reports on the results of a survey sent to libraries in US and Canada and a literature search regarding libraries and librarians’ contribution to walking for health.

Introduction

The librarians are not contributing to a walkable neighborhood but rather are providing opportunities for walking in the neighborhood.  I suggest changing this wording throughout the paper.

Suggest deleting the sentence on lines 91-92, The discussion section suggests additional research needed 92 to better understand this topic, and to bring public librarianship and public health closer together, as it out of place in the introduction.

Methods

More information is needed on the survey, how many questions were on the survey, what other types of questions were on the survey. For example, the size (number of employees) of the library, size of the community served, resources available to library?

Was the survey sent to all 19,564 public library locations and only 1,157 returned a completed survey for a response rate of 5.9%? Was an email with survey link sent out once was a second email sent to non- responders? Was the survey sent to one librarian per library?

It is not clear how the authors ended up with 94 publications.  I am not sure how publications are added to the “corpus of online publications collected on active living programs in public libraries”.  What types of publications are included in this corpus? From reference lists looks like it includes posting on social media or blogs. Were publications that are not published online (e.g. print journals) not included in this collection? And if so how does that limit the information that this review is based upon. More detail is needed regarding this corpus of publications. Suggest include a table that lists the types/sources and number per type/source of each publication Did the authors conduct a literature search? If so more information is needed on methods of the search

More detail is needed on how the 94 articles were analyzed.  From the superficial description in the methods and the presentation of description of results this does not appear to a grounded theory study but appears to be possibly a qualitative descriptive study or qualitative content analysis.

Results:

The 482 libraries that reported offering some type of walking program is 42 % of those who responded to the survey. To state that it is 2.5% of all libraries in US and Canada is not accurate since it is not known whether the 18,407 libraries that did not return a survey offer walking programs.  The percentages for story walks and walking and biking in general are also not based on respondents.  The percentages in this paragraph need to be changed to reflect percent of respondents not total libraries in US and Canada.

The quotes in the results are these direct quotes of individuals quoted in the cited source? If so should include page number. This needs to be clarified.

Suggest adding a table of the survey results that reports on the number of libraries that offer each type of program as well as the size of the community served by the library or size of the library and other pertinent information regarding the libraries asked in the survey.

Discussion

Need to comment on sources included and limitations to those sources as well as limitations to search strategy used.

Author Response

Point 1: The librarians are not contributing to a walkable neighborhood but rather are providing opportunities for walking in the neighborhood.  I suggest changing this wording throughout the paper.

Response 1: This change has been made throughout the manuscript.

Point 2: Suggest deleting the sentence on lines 91-92, The discussion section suggests additional research needed 92 to better understand this topic, and to bring public librarianship and public health closer together, as it out of place in the introduction.

Response 2: This change has been made.

Point 3: More information is needed on the survey, how many questions were on the survey, what other types of questions were on the survey. For example, the size (number of employees) of the library, size of the community served, resources available to library?

Response 3: The survey instrument has been included as an appendix, and a description of the survey and its questions has been inserted into the text.

Point 4: Was the survey sent to all 19,564 public library locations and only 1,157 returned a completed survey for a response rate of 5.9%? Was an email with survey link sent out once was a second email sent to non- responders? Was the survey sent to one librarian per library?

Response 4: Clarification around the survey method added to text, focusing on the fact that the survey consisted of a self-selecting sample of libraries and was not sent to individual libraries.

Point 5: It is not clear how the authors ended up with 94 publications.  I am not sure how publications are added to the “corpus of online publications collected on active living programs in public libraries”.  What types of publications are included in this corpus? From reference lists looks like it includes posting on social media or blogs. Were publications that are not published online (e.g. print journals) not included in this collection? And if so how does that limit the information that this review is based upon. More detail is needed regarding this corpus of publications. Suggest include a table that lists the types/sources and number per type/source of each publication Did the authors conduct a literature search? If so more information is needed on methods of the search

Response 5: Table 2 has been added to provide the reader with more information on this corpus. A sentence has been added to the discussion section outlining the limitations (as well as the strengths) of this approach. More generally, the methods section includes a larger discussion of content analysis, and the strengths of qualitative methods for exploratory research.

Point 6: More detail is needed on how the 94 articles were analyzed.  From the superficial description in the methods and the presentation of description of results this does not appear to a grounded theory study but appears to be possibly a qualitative descriptive study or qualitative content analysis.

Response 6: More clarification about content analysis as a method has been introduced into the text to clarify how analysis proceeded.

Point 7: The 482 libraries that reported offering some type of walking program is 42 % of those who responded to the survey. To state that it is 2.5% of all libraries in US and Canada is not accurate since it is not known whether the 18,407 libraries that did not return a survey offer walking programs.  The percentages for story walks and walking and biking in general are also not based on respondents.  The percentages in this paragraph need to be changed to reflect percent of respondents not total libraries in US and Canada.

Response 7: This change has been made – percentages now refer to percentages of respondents.

Point 8: The quotes in the results are these direct quotes of individuals quoted in the cited source? If so should include page number. This needs to be clarified.

Response 8: Page numbers (and paragraph numbers, where no page number is available) have been added to direct quotes in the text.

Point 9: Suggest adding a table of the survey results that reports on the number of libraries that offer each type of program as well as the size of the community served by the library or size of the library and other pertinent information regarding the libraries asked in the survey.

Response 9: Table 1 has been added with this information.

Point 10: Need to comment on sources included and limitations to those sources as well as limitations to search strategy used.

Response 10: This has been added to the discussion section.

Reviewer 2 Report

I think this paper is of great interest and raises many interesting questions. Although it is primarily qualitative in nature, I think that your argument that this needs to be further researched would be supported by further details about the quantitative data you describe RE: survey. In particular, 482 out of 1157 libraries indicated that they offered walking programs - how do these libraries differ from those that don't? Demographics, location, city characteristics, circulation... this seems like it would be fairly easy to add and would contribute substantially to your paper.

Otherwise, this paper needs a thorough edit. There are many grammatical/typographical errors.

Author Response

Point 1: I think this paper is of great interest and raises many interesting questions. Although it is primarily qualitative in nature, I think that your argument that this needs to be further researched would be supported by further details about the quantitative data you describe RE: survey. In particular, 482 out of 1157 libraries indicated that they offered walking programs - how do these libraries differ from those that don't? Demographics, location, city characteristics, circulation... this seems like it would be fairly easy to add and would contribute substantially to your paper.

Response 1: Table 1 has been added to address this issue. Thank you for the suggestion.  

Point 2: Otherwise, this paper needs a thorough edit. There are many grammatical/typographical errors.

Response 2: A thorough copy-editing has been performed. My apologies for these errors.